# Gyre Precoding and T-Transformation-Based GFDM System for UAV-Aided mMTC Network

**Joarder Jafor Sadique** [1,*] , **Shaikh Enayet Ullah** [2] , **Raad Raad** [3] , **Md. Rabiul Islam** [3] , **Md. Mahbubar Rahman** [4] , **Abbas Z. Kouzani** [5] and **M. A. Parvez Mahmud** [5]

1   Department of Electrical and Electronic Engineering, Begum Rokeya University, Rangpur 5404, Bangladesh
2   Department of Electrical and Electronic Engineering, University of Rajshahi, Rajshahi 6205, Bangladesh; enayet_apee@ru.ac.bd
3   Faculty of Engineering and Information Sciences, University of Wollongong, Wollongong, NSW 2522, Australia; raad@uow.edu.au (R.R.); mrislam@uow.edu.au (M.R.I.)
4   Department of Electrical and Electronic Engineering, Islamic University, Kushtia 7003, Bangladesh; mahbublv@eee.iu.ac.bd
5   School of Engineering, Deakin University, Geelong, VIC 3216, Australia; abbas.kouzani@deakin.edu.au (A.Z.K.); m.a.mahmud@deakin.edu.au (M.A.P.M.)
*   Correspondence: joarder@brur.ac.bd

**Abstract:** In this paper, an unmanned aerial vehicle (UAV)-aided multi-antenna configured downlink mmWave cooperative generalized frequency division multiplexing (GFDM) system is proposed. To provide physical layer security (PLS), a 3D controlled Lorenz mapping system is introduced. Furthermore, the combination of T-transformation spreading codes, walsh Hadamard transform, and discrete Fourier transform (DFT) techniques are integrated with a novel linear multi-user multiple-input multiple-output (MU-MIMO) gyre precoding (GP) for multi-user interference reduction. Furthermore, concatenated channel-coding with multi-user beamforming weighting-aided maximum-likelihood and zero forcing (ZF) signal detection schemes for an improved bit error rate (BER) are also used. The system is then simulated with a single base station (BS), eight massive machine-type communications (mMTC) users, and two UAV relay stations (RSs). Numerical results reveal the robustness of the proposed system in terms of PLS and an achievable ergodic rate with signal-to-interference-plus-noise ratio (SINR) under the implementation of T-transformation scheme. By incorporating the 3D mobility model, brownian perturbations of the UAVs are also analyzed. An out-of-band (OOB) reduction of 320 dB with an improved BER of $1 \times 10^{-4}$ in 16-QAM for a signal-to-noise ratio, $E_b/N_0$, of 20 dB is achieved.

**Keywords:** generalized frequency division multiplexing; cooperative unmanned aerial vehicle; massive machine type communication; physical layer security; T-transformation spreading codes; out-of-band; signal-to-interference-plus-noise ratio; gyre precoding

## 1. Introduction

An uncrewed aircraft handled by remote control or embedded computer programs is commonly known as an unmanned aerial vehicle (UAV) or drone. Recently, there has been tremendous amount of interest growing to develop UAV-ground communications using low-cost massive UAVs under existing 5G as well as future-generation (B5G/6G) cellular networks. To maintain secure and reliable flight operation, UAVs can exchange safety-critical information with remote pilots, closest aerial vehicles, and air traffic controllers with an assistance of control and non-payload communication (CNPC). UAVs are also delivering goods and improving the throughput of 5G networks [1–3].

Additionally, mission-oriented data, such as high-resolution video, data packets, and aerial images, are possible to seamlessly transfer with UAVs in payload communication. Even in an emergency situation when terrestrial mobile stations go through unexpected discontinuity due to disasters, UAVs can be utilized as a base station (BS) or as a relay

station (RS). Due to flexible deployment and cost effectiveness, full-duplex UAVs can also be integrated with terrestrial cellular networks [4]. A typical scenario is illustrated in Figure 1, where UAVs are forming a cooperative relay network with an assumption that there is no direct connection between BS and user equipment (UE).

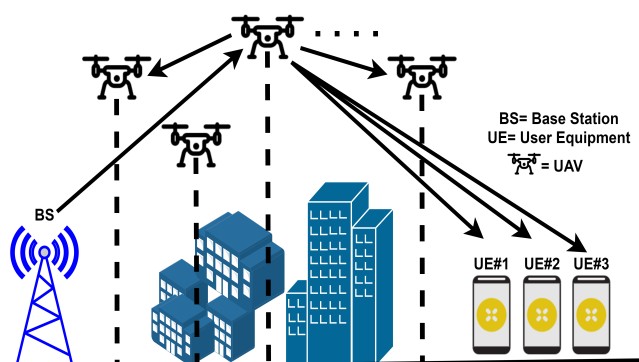

**Figure 1.** A cooperative UAV-enabled relay network.

The major difference between CNPC and UAV payload communication is that it requires a much higher data rate to initiate the transfer of full high-definition (FHD) video from the UAV to the ground user later [5]. However, it is anticipated that the commercially deployed 5G network would provide more diverse services that include massive machine-type communications (mMTC), enhanced mobile broadband (eMBB), and ultra-reliable low-latency communications (URLLC) [6,7].

The crucial challenge for the cellular network of the upcoming generation would be ensuring the privacy of the communications messages. The presence of unwanted users, or more specifically known as eavesdropper, always makes the exchange of information between the transmitting and receiving end vulnerable to malicious attacks due to the openness of wireless communication systems. To safeguard data transmission over the communication network, physical layer security (PLS) has emerged as an effective solution to such confidentiality issues in wireless connections [8]. Sustaining proper encryption at the physical layer can efficiently increase the security level to high, and eventually, it will make the whole system more reliable and secure.

Compared to other practical techniques, chaos-based encryption has become a prominent process to maintain high-level of efficacy as far as the security is concerned. In such technique, stealing of information becomes tremendously difficult for intruding users as the transmitter and receiver share an initial value exclusively between them. The Lorenz mapping system is such a chaotic-based encryption scheme that enables a completely unpredictable numerical sequence of system variables. Furthermore, the addition of partial control variables to the design process in Lorenz's hyperchaos mapping system enhances the keyspace of the encryption algorithm [9,10].

To provide high-quality eMBB services considering the robustness of the terrestrial network capacity and alleviate the effect of network congestion or network failure, a multi-UAV-aided non-orthogonal multiple access (NOMA) network is preferred in [11]. Such multicarrier NOMA technology-based UAV-terrestrial integrated network is considered for supporting fast communication service recovery or even enhancing the network performance. The authors of [12] provided an extensive review study on the evolution of non-terrestrial networks (NTNs) and highlighted partially and fully integrated ground-air-space (GAS) networks from 5G to 6G by discussing the techniques ranging from new services (e.g., Internet of things (IoT) and multi-access edge computing (MEC)), to new spectrum bands (e.g., mmWave and THz), and to new approaches (e.g., machine learning (ML)). In [13], a global energy-efficient ant colony optimization (ACO) routing algorithm for UAV-aided mMTC networks is proposed to minimize the energy consumption and extend the lifetime of the networks. Furthermore, a practical algorithm is proposed to reduce

the complexity. In [14], a orthogonal frequency division multiplexing (OFDM) combined index modulation signaling scheme is incorporated with UAV communication systems and results in higher spectral efficiency, higher energy efficiency, and lower BER. Generally, the OFDM system is associated with drawbacks related to issues such as high peak-to-average power ratios (PAPR), bandwidth loss associated with the cyclic prefix (CP), and high out-of-band (OOB) emissions. To meet the challenges of fifth generation (5G) and B5G wireless communication networks to handle peak data rates of at least 1 Tb/s, over-the-air latency of 10–100 µs, and user-experienced data rates of 1 Gb/s, generalized frequency division multiplexing (GFDM), non-orthogonal multicarrier modulation (MCM) scheme may be preferred in lieu of OFDM. The future generation wireless networks should target ensuring low power consumption for machine-to-machine communication and low latency for tactile internet and internet-of-things (IoT) [15]. By reducing inter-carrier interference (ICI) and inter-symbol interference (ISI) through a properly designed better than raised cosine (BTRC) pulse shaping filter, improved sinc power (ISP), parametric linear pulse (PLP), and parametric exponential (PEXP), bit error rate (BER) performance is improved in the GFDM system [16]. In [17], a complementary GFDM repeated transmission scheme is proposed to enhance BER performance for mMTC and IoT. A system model is presented by the authors of [18] to develop an Einstein product of a tensor-based precoding scheme suitable for MIMO GFDM systems. In [19], a joint data symbol detection and phase noise compensation algorithm is proposed to reduce the effects of phase noise in GFDM systems. In this present study, a cooperative UAV-aided millimeter wave (mmWave) CP-free GFDM system is proposed considering data transmission for ground mMTC. In broadband wireless communication systems, one of the most popular modulation schemes, CP-OFDM, utilizes the CP to make the frequency-selective multipath fading channel appear as flat-fading [20]. As the CP causes overhead in the spectrum efficiency and power efficiency, the use of a CP is being avoided, and guard intervals (GIs) are being introduced between two consecutive GFDM symbols. In this UAV relay-assisted terrestrial networking system, more focus is attributed to enhancing secure transmission using Lorenz's hyperchaos mapping system addressed in [10].

This paper mainly deals with a multi-antenna-configured UAV-aided cooperative CP-less GFDM system for secured B5G wireless networks. The physical size of each antenna of the multi-antenna-mounted UAV is very small since mmWave transmission frequency (28 GHz) is considered for this scheme. Generally, MIMO antenna-configured systems are preferably used to enhance the diversity gain and data rate of the system concerned. As the total transmission power of the UAV is limited, a single power-amplifying section for each transmitting layer is utilized here instead of using large RF chains, and thus, any negative effect on operational life cycle of the UAV is being minimized.

NOMA, GFDM, universal filtered multicarrier (UFMC), and filter bank multicarrier (FBMC) has always been regarded as an excellent contender waveform for 5G networks. Among the mentioned, GFDM is a symbol-based multicarrier multiplexing technique which keeps the signal well confined in time and frequency domains. It utilizes circular filter for effective reduction in the low out-of-band (OOB) emission and ensures simultaneous transmission of multiple symbols at different time slots. By taking all these properties into account, a GFDM-aided system is presented in place of the OFDM-based signaling technique. As far as the novelty of this work is concerned, designing a system for the mMTC network utilizing a CP-less UAV-aided GFDM system has not been well investigated before. Significant contributions of the paper are outlined as follows:

- T-transformation spreading codes in combination with the walsh Hadamard transform (WHT) and discrete Fourier transform (DFT) is proposed along with recently the introduced gyre precoding (GP) [21] technique. This reduces multi-user interference (MUI) and computational complexity.
- The BTRC pulse shaping filter and concatenated channel-coding techniques are utilized to reduce OOB power emission and improve BER performance.

- Shen et al.'s proposed a 3D controlled Lorenz mapping system [10] is implemented as a cryptographic technique in this proposed system for PLS encryption.
- The 3D mobility model is also introduced to observe 3D positional uncertainty of the UAVs due to Brownian motion and analyzed its impact on the achievable ergodic rate.
- Simulation results verify the significant enhancement of the achievable ergodic rate, bit error rate (BER) performance, OOB reduction at a reasonably acceptable PAPR.

The structure of this paper is as follows. Section 2 describes the system model, including the block diagram and insights of the signal model considered. In Section 3, simulation parameters, numerical results, and outcomes of the proposed system are presented. Finally, a brief discussion on final remarks and future directions are provided in Section 4.

## 2. System Model

### 2.1. Block Diagram

The conceptual block diagram of the secured cooperative UAV-based CP-less GFDM system is shown in Figure 2. In its ground segment section, initial significant signal processing operations are executed on the ground BS. Ground BS is set to be comprised of $N_T$ (=16) transmitting antennas, which should serve eight ground mMTC users via one main UAV and two cooperative UAVs, and all the UAVs are $N_T \times N_T$ multi-antenna configured. The synthetically generated binary data assigned for individual mMTC users are processed with concatenated channel-coding schemes based on (3,2) single parity check (SPC), low density parity check (LDPC), and repeat and accumulate (RA) [22,23] and subsequently encrypted with improved 3D controlled Lorenz mapping-based encryption technique [10].

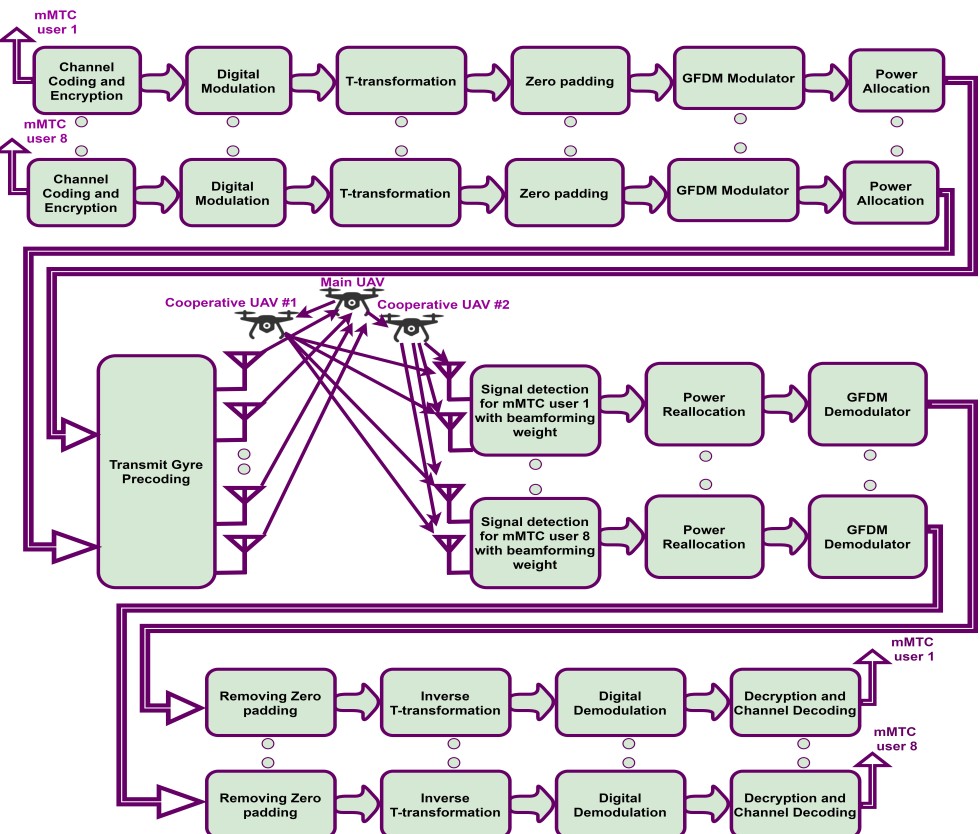

**Figure 2.** Simplified block diagram of secured cooperative UAV-aided CP-less GFDM system.

Initially, complex symbols are produced via the digital modulation of the encrypted binary data [24]. After that, the generated complex symbols undergo T-transformation with a combined effect of the WHT and DFT [25] and zero padded prior to the execution

of the GFDM modulation. In the GFDM modulation, the zero padded complex symbols are circularly convolved with the BTRC filter, followed by frequency up conversion [16,26]. The GFDM modulated symbols are power scaled and subsequently precoded with channel-dependent multi-user transmit GP schemes [21]. The transmission of the spatially multiplexed precoded signals from the ground BS to the main UAV with ending signal processing operations in the ground segment section of this proposed system.

In the UAV segment, the main UAV decodes the transmitted signal with the ZF linear signal detection technique [27], and the power of the detected signal is boosted up to make it compatible with the power of the ground BS. In the first time phase, the main UAV transmits the detected signal simultaneously to the cooperative UAVs working in half-duplex mode and the eight ground mMTC users, each of which is equipped with $N_R$ (=2) receiving antennas. In the second time phase, the cooperative UAVs simply adopt the amplify-and-forward (AF) relaying strategy [28] and forwards the signal to the ground mMTC users with identical transmission power of the main UAV. On the basis of the estimated effective channel between the UAVs and the individual mMTC user in two consecutive time phases and their respective beamforming weight, mMTC user's own transmitted signal is detected at the receiving end of each mMTC user [29]. The detected signals are power scaled again to be restored at the original power. The GFDM demodulation process is then initiated by the removal of the padded zeros, and the multiplication with inverse T-transformation process yields despreaded signals. Despreaded signals are then passed through the phases of digital demodulation, channel decoding, and improved 3D controlled Lorenz mapping-based decryption to eventually recover the transmitted data.

### 2.2. Signal Model

In the previous section, various signal processing techniques have been addressed by presenting their applicability in this proposed system. In this section, a significant emphasis has been put on the signal model to describe various useful processing techniques. This can be divided in two segments first, which are: (1) the Ground Segment and (2) the UAV Segment.

### 2.2.1. Ground Segment

To improve both transmission and the performance in terms of security, the 3D controlled Lorenz mapping system addressed in [10] can be represented as:

$$\left.\begin{array}{ll} \dot{x} & = \rho(y - x) + \zeta x mod(0.001, \mu) \\ \dot{y} & = rx - y - xz + \zeta y mod(0.001, \mu) \\ \dot{z} & = xy - \beta z + \zeta z mod(0.001, \mu) \end{array}\right\} \tag{1}$$

where the initial values of and $x, y, z$ variables and key parameter values $(x, y, z, \rho, \zeta, r, \mu, \beta)$ have been assigned to $(1.99, 2.2, 2.0, 9, 2, 35, 25, 8/3)$. The solution component $\{z(t)\}_{t=1}^{768}$ can be used to produce $768 \times 1$ matrix-sized primary key $K_0$ as:

$$K_0 = \lceil \{z(t)\}_{t=1}^{768} \rceil \tag{2}$$

where $\lceil \cdot \rceil$ is indicative of rounding operation to the nearest integer. The primary key $K_0$ is disseminated into eight encrypted keys with each key containing 96 elements. The eight keys for the eight mMTC users can be considered as:

$$\left.\begin{array}{ll} K_1 & = \lceil 0.5 K_0(1:96, 1) \rceil \\ K_2 & = \lceil 1.0 K_0(97:192, 1) \rceil \\ K_3 & = \lceil 1.5 K_0(193:288, 1) \rceil \\ K_4 & = \lceil 2.0 K_0(289:384, 1) \rceil \\ K_5 & = \lceil 2.5 K_0(385:480, 1) \rceil \\ K_6 & = \lceil 3.0 K_0(481:576, 1) \rceil \\ K_7 & = \lceil 3.5 K_0(577:672, 1) \rceil \\ K_8 & = \lceil 4.0 K_0(673:768, 1) \rceil \end{array}\right\} \tag{3}$$

where the first to 96th elements of $K_0$ is designated by $K_0(1:96,1)$, 97th to 192th elements of $K_0$ is designated by $K_0(97:192,1)$, and so on. The elemental values of all the keys $K_1$ through $K_8$ fall within the range, and each elemental value provides eight binary bits. In binary form, each of the generated keys $K_1$ through $K_8$ for eight mMTC users is of $1 \times 768$ matrix in size. By using the repmat function in MATLAB, the length of the binary data of each of the generated keys $K_1$ through $K_8$ is made identical to the data length of the concatenated channel encoded data. The encrypted concatenated channel encoded binary data vector $b_{\bar{k}}$ of data length $\bar{N}$ for mMTC user $\bar{k}$ can be represented as:

$$\bar{b}_{\bar{k}} = b_{\bar{k}} \oplus \bar{K}_{\bar{k}} \tag{4}$$

where the symbol $\oplus$ is indicative of the XOR operation, and $\bar{K}_{\bar{k}}$ is the ultimately generated key for the mMTC user $\bar{k}$. With the data length $\tilde{N}$, the conversion of binary data vector $\bar{b}_{\bar{k}}$ as a complex symbol vector yields $c_{\bar{k}}$ through digital modulation. The T-transformed $8 \times 8$ sized matrix based on combining the WHT and DFT can be represented as:

$$T = \frac{1}{\sqrt{8}} \begin{bmatrix} 1 & 1 & 1 & 1 & 1 & 1 & 1 & 1 \\ 1 & \omega^{-1} & \omega^{-2} & \omega^{-3} & -1 & \omega^{-1} & \omega^{-2} & \omega^{-3} \\ 1 & \omega^{-2} & -1 & -\omega^{-2} & 1 & \omega^{-2} & -1 & -\omega^{-2} \\ 1 & \omega^{-3} & -\omega^{-2} & \omega^{-1} & -1 & -\omega^{-3} & \omega^{-2} & -\omega^{-1} \\ 1 & -1 & 1 & -1 & 1 & -1 & 1 & -1 \\ 1 & -\omega^{-1} & \omega^{-2} & -\omega^{-3} & -1 & \omega^{-1} & -\omega^{-2} & \omega^{-3} \\ 1 & -\omega^{-2} & -1 & \omega^{-2} & 1 & -\omega^{-2} & -1 & \omega^{-2} \\ 1 & -\omega^{-3} & -\omega^{-2} & -\omega^{-1} & -1 & -\omega^{-3} & \omega^{-2} & \omega^{-1} \end{bmatrix} \tag{5}$$

where $\omega = e^{j2\pi/8}$. The signal model presented in Equation (5) is applicable row wise for different mMTC users $\bar{k}$. The first row is applicable to mMTC user 1, the second row is applicable to mMTC user 2, and so on. The application of the T-transformation spreading technique on the modulated complex symbol vector $c_{\bar{k}}$ for mMTC user $\bar{k}$ yields the complex symbol vector $\ddot{c}_{\bar{k}}$ of data length $8\tilde{N}$.

In GFDM structured block generation, the data vector $\ddot{c}_{\bar{k}}$ is rearranged into a $K \times M$ matrix-sized GFDM block, where $M$ represents the number of subsymbols and $K$ denotes the number of subcarriers. For every GFDM block with N(=KM) samples, its first and last subsymbol contain null subcarriers to avoid collision in between two consecutive GFDM blocks, eliminate the effect of the fading channel, and additionally provide significant OOB emissions reduction for neighboring channel interference reduction. Considering data symbol $d_{k,m,l}$ in each GFDM block for mMTC user $\bar{k}$, the $l$th GFDM block for mMTC user $\bar{k}$ can be seen as a result of circular convolution with BTRC, $g[n]$, followed by frequency up conversion and can be written as [26]:

$$x_{\bar{k},l}[n] = \sum_{k=0}^{K-1} \left[ g(n) \circledast \sum_{m=0}^{M-1} d_{k,m,l} \delta[n-mK] \right] e^{j2\pi \frac{k}{K} n} \tag{6}$$

where $\circledast$ denotes circular convolution, $\bar{k} = 1,2,\ldots,8$ is the mMTC user identification number, $l = 1,2,\ldots,L$ is the GFDM block identification number, $\delta[n-mK]$ is the Dirac function with $n = 0,1,2,\ldots,N-1$, and $m = 0,1,2,\ldots,M-1$.

The BTRC pulse shaping filter addressed in [21] and used in signal model of Equation (5) can be written in the time domain as:

$$g(t) = \frac{1}{T} \operatorname{sinc} \frac{t}{T} \times \frac{4\beta t \sin\left(\frac{\pi \alpha t}{T}\right) + 2\beta^2 \cos\left(\frac{\pi \alpha t}{T}\right) - \beta^2}{\beta^2 + (2\pi t)^2} \tag{7}$$

where $\beta = \frac{ln2}{\alpha B}$, $T\left(= M\Delta T = \frac{N}{F_s}\right)$ is the GFDM symbol period, $\alpha$ is the filter roll-off factor of the BTRC pulse shaping filter, $\Delta T\left(= \frac{K}{F_s}\right)$ is the period of each subsymbol, $F_s$ is the sampling frequency, $\Delta f\left(= \frac{1}{\Delta T}\right)$ represents the subcarrier spacing, and $B\left(= K\Delta f = \frac{K}{\Delta T}\right)$ represents the GFDM signal bandwidth. The $l$th discrete time domain GFDM signal (Equation (6)) for mMTC user $\bar{k}$ is rescaled such that $E|x_{\bar{k},l}|^2 = 1$, and the power rescaled signal vector of unity power is represented by $\tilde{x}_{\bar{k},l}[n]$. Concatenation of all the signal vectors of $\tilde{x}_{\bar{k},l}[n]$ yields the desired signals for the mMTC user $\bar{k}$ and are denoted by a data matrix $D_{\bar{k}}$ of size $N \times L$. All the elements of $D_{\bar{k}}$ are stacked within a single

column vector and further arranged into a matrix $\bar{D}_{\bar{k}}$ of $N_R \times N_L (= N \times L)/N_R$ in size, and the $\bar{D}_{\bar{k}}$ can be represented as:

$$\bar{D}_{\bar{k}} = \ddot{x}_{\bar{k}} \tag{8}$$

From Equation (8), all the data for eight mMTC users can be confined into a single data matrix D of size $N_T \times N_L$.

Prior to the application of proper transmit beamforming weights to the matrixed data, D, it is desirable to design transmit precoders utilizing the 3D geometrical MIMO flat-fading channels, which are based on the probabilistic path loss model, and in such estimated fading channels, the 3D mobility model has been incorporated to consider 3D positional uncertainty of the main UAV and cooperative UAV's positions relative to their fixed positions due to Brownian motion. The 3D mobility model can be used to present positioning errors using homogeneous stochastic differential equations in the 3D spatial $(x, y, z)$ coordinate system as:

$$\left.\begin{array}{l} dx_{t,s} = -\alpha_1 \tilde{x}_{t,s} dt + \sigma_1 dw_{1t} \\ dy_{t,s} = -\alpha_2 \tilde{y}_{t,s} dt + \sigma_2 dw_{2t} \\ dz_{t,s} = -\alpha_3 \tilde{z}_{t,s} dt + \sigma_3 dw_{3t} \end{array}\right\} \tag{9}$$

where $\alpha_1, \alpha_2, \alpha_3, \sigma_1, \sigma_2, \sigma_3$ are the six constants ($\alpha_1 = \alpha_2 = \alpha_3 = 1$ and $\sigma_1 = 1.3, \sigma_2 = 1.0, \sigma_3 = 0.7$); $dw_{1t}, dw_{2t}, dw_{3t}$ are the differential form of the Brownian motion in three mutually perpendicular $x$, $y$, and $z$ directions and are equally applicable for all UAVs. In Brownian motion, it can be assumed that $[dw_{1t}]^2 = [dw_{2t}]^2 = [dw_{3t}]^2 = dt$. The $(x_{t,s}, y_{t,s}, z_{t,s})$ are the 3D positional coordinates of the s-th UAV ($s = 0$ for main UAV, $s = 1$ for cooperative UAV #1 and $s = 2$ for cooperative UAV #2). The $(\tilde{x}_{t,s}, \tilde{y}_{t,s} \tilde{z}_{t,s})$ are the 3D errored positional coordinates of the s-th UAV due to Brownian motion [30,31]. With the consideration of Brownian motion, the $x$, $y$, and $z$ coordinates of the main UAV and the cooperative first and second UAVs are $(\tilde{x}_{t,0}, \tilde{y}_{t,0} \tilde{z}_{t,0})$, $(\tilde{x}_{t,1}, \tilde{y}_{t,1} \tilde{z}_{t,1})$, and $(\tilde{x}_{t,2}, \tilde{y}_{t,2} \tilde{z}_{t,2})$, respectively. The 3D distance $d_{\overline{k,j}}$ between main/cooperative UAV $\bar{j}$ and the ground mMTC user $\bar{k}$ with its 3D coordinates $(x_{\bar{k}}, y_{\bar{k}}, z_{\bar{k}})$ can be represented as: $d_{\overline{k,j}} = \sqrt{(\tilde{x}_{t,s} - x_{\bar{k}})^2 + (\tilde{y}_{t,s} - y_{\bar{k}})^2 + (\tilde{z}_{t,s} - h_{\bar{k}})^2}$, and the elevation angle $\theta_{\overline{k,j}}$ can be represented as: $\theta_{\overline{k,j}} = \frac{180}{\pi} \times \sin^{-1} \frac{\tilde{z}_{t,s} - h_{\bar{k}}}{d_{\overline{k,j}}}$. The LOS probability from UAVs to ground mMTC communications can be represented as [32]:

$$P_{LOS}^{\overline{k,j}} = \frac{1}{1 + \psi exp[-\beta_0(\theta_{\overline{k,j}} - \psi)]} \tag{10}$$

where $s = 0, 1, 2$, and $\psi$ (=11.95) and $\beta_0$ (=0.14) are the functions of the carrier frequency and environment. The path loss between the main/cooperative UAV $\bar{j}$ and the ground mMTC user $\bar{k}$ can be represented as:

$$L_{\overline{k,j}} = \left\{ \begin{array}{ll} \eta_1 \left(\frac{4\pi f_c d_{\overline{k,j}}}{c}\right)^\alpha, & \text{LOS Link} \\ \eta_2 \left(\frac{4\pi f_c d_{\overline{k,j}}}{c}\right)^\alpha, & \text{NLOS Link} \end{array} \right\} \tag{11}$$

where $\alpha$ is the path loss exponent, $f_c$ (=28 GHz) is the carrier frequency, $\eta_1$ (=$10^{3/10}$) represents the excessive path loss coefficient for LOS link, $\eta_2$ (=$10^{23/10}$) denotes the excessive path loss coefficient for NLOS link, and $c$ (=$3 \times 10^8$) m/s represents the speed of light. The simple form of NLOS probability is $P_{NLOS}^{\overline{k,j}} = 1 - P_{LOS}^{\overline{k,j}}$. Between the main/cooperative UAVs $\bar{j}$ and the ground mMTC user $\bar{k}$, the average path loss equation can be represented as:

$$\bar{L}_{\overline{k,j}} = P_{LOS}^{\overline{k,j}} \eta_1 \left(\frac{4\pi f_c d_{\overline{k,j}}}{c}\right)^\alpha + P_{NLOS}^{\overline{k,j}} \eta_2 \left(\frac{4\pi f_c d_{\overline{k,j}}}{c}\right)^\alpha \tag{12}$$

With its 3D coordinates $[x_0, y_o, z_o]$ to the main UAV, the path loss $L_0$ of the MIMO flat-fading channel $H_0$ for the case of uplink transmission from ground BS can be represented as:

$$L_0 = P_{LOS}^{\overline{g,m}} \eta_1 \left(\frac{4\pi f_c d_{\overline{g,m}}}{c}\right)^\alpha \tag{13}$$

where $d_{\overline{g,m}} = \sqrt{(\tilde{x}_{t,0} - x_0)^2 + (\tilde{y}_{t,0} - y_0)^2 + (\tilde{z}_{t,0} - z_0)^2}$ and $P_{LOS}^{\overline{g,m}} = \frac{1}{1 + \psi exp\left[-\beta_0\left(\frac{180}{\pi} \times \sin^{-1} \frac{\tilde{z}_{t,0} - z_0}{d_{\overline{g,m}}} - \psi\right)\right]}$.

Considering very high LOS probability and a low excessive path loss for transmission between the main UAV and the cooperative UAVs, the path losses $L_{01}$ and $L_{02}$ for the MIMO flat-fading channels $H_{01}$ and $H_{01}$ existed between the main UAV and first UAV, and the main UAV and second UAV can be represented as:

$$L_{01} = \eta_1 \left( \frac{4\pi f_c d_{\overline{1,m}}}{c} \right)^{\alpha} \tag{14a}$$

$$L_{02} = \eta_1 \left( \frac{4\pi f_c d_{\overline{2,m}}}{c} \right)^{\alpha} \tag{14b}$$

where $d_{\overline{1,m}}$ and $d_{\overline{2,m}}$ are the corresponding 3D distances between the main UAV and cooperative UAVs. Considering the applicability of the signal models represented in Equations (11)–(14b), the MIMO flat-fading channels from the main UAV to eight mMTC users (H1, H2, H3, H4, H5, H6, H7, H8), from the first cooperative UAV to eight mMTC users (H11, H12, H13, H14, H15, H16, H17, H18), from the second cooperative UAV to eight mMTC users (H21, H22, H23, H24, H25, H26, H27, H28), H0, $H_{01}$, and $H_{02}$ are estimated. The total MIMO flat-fading channel matrix H of size $N_T \times N_T$ for all the mMTC users can be formulated as:

$$H = [H_1^T \ H_2^T \ H_3^T \ H_4^T \ H_5^T \ H_6^T \ H_7^T \ H_8^T]^T \tag{15}$$

where $(\cdot)^T$ indicates the non-conjugate transpose operator. Considering the GP, a novel linear multi-user multiple-input multiple-output (MU-MIMO) precoding scheme cited at [21], the ZF precoding of matrix size $N_T \times N_T$ can be represented as:

$$G^{(ZF)} = H^H (HH^H)^{-1} \tag{16}$$

where $(\cdot)^H$ indicates the conjugate transpose operator.

In data matrix D, its first and second rowed data are belonging to first mMTC user, third and fourth rowed data are belonging to second mMTC user, and so on. To estimate GP, $W^{(GP)}$, two additional components $\Theta$ and $\gamma^{GP}$ are in need of computation. The rotation angle matrix $\Theta$ can be represented as:

$$\Theta = \begin{bmatrix} e^{j\theta_{11}} & 0 & 0 & \cdots & \cdots & \cdots & \cdots & 0 \\ 0 & e^{j\theta_{11}} & \cdots & & 0 & \cdots & & 0 \\ & & e^{j\theta_{22}} & & & & & \\ 0 & & 0 & e^{j\theta_{22}} & & & & 0 \\ & & & \vdots & & & & \\ & & 0 & & & & e^{j\theta_{16}} & \\ & 0 & & & & & & e^{j\theta_{16}} \end{bmatrix} \tag{17}$$

where $\theta_{\overline{kk}} \epsilon [0, 2\pi]$ and

$$\gamma^{GP} = \frac{1}{NL} ||G^{(ZF)} \Theta D||_F^2 \tag{18}$$

where $||\cdot||_F$ is the Frobenius norm. The GP matrix $W^{(GP)}$ is of size $N_T \times N_T$ and can be represented as:

$$W^{(GP)} = \frac{G^{(ZF)} \Theta}{\sqrt{\gamma^{GP}}} \tag{19}$$

The precoding weights W1 through of W8, each is of $N_T N_R$ in size for the eight mMTC users and can be represented as: $W_1 = W^{(GP)}(:, 1 : 2); W_2 = W^{(GP)}(:, 3 : 4); W_3 = W^{(GP)}(:, 5 : 6); W_4 = W^{(GP)}(:, 7 : 8); W_5 = W^{(GP)}(:, 9 : 10); W_6 = W^{(GP)}(:, 11 : 12); W_7 = W^{(GP)}(:, 13 : 14);$ and $W_8 = W^{(GP)}(:, 15 : 16)$, where $W^{(GP)}(:, 1 : 2)$ is indicative of first and second columned data of GP matrix $W^{(GP)}$ and so on. Precoded signal $X_0$ trasmitted from ground BS can be represented as:

$$X_0 = W^{(GP)} D = W_{\bar{k}} \ddot{x}_{\bar{k}} + \sum_{\bar{j}=1, \bar{j} \neq \bar{k}}^{\bar{j}=8} W_{\bar{j}} \ddot{x}_{\bar{j}} \tag{20}$$

### 2.2.2. UAV Segment

Due to precoding, signal power varies from rowed to rowed data of $X_0$, and to make identical signal power in every rows of $X_0$, it is to be multiplied by a normalized signal power $P_1$ such that the transmitted power from the ground BS would be $P_{gr}$ and the arrived signal at the main UAV can be represented as:

$$Y_0 = H_0 \sqrt{P_1} \left\{ W_{\bar{k}} \ddot{x}_{\bar{k}} + \sum_{\bar{j}=1, \bar{j} \neq \bar{k}}^{\bar{j}=8} W_{\bar{j}} \ddot{x}_{\bar{j}} \right\} + n_m \tag{21}$$

where $n_m \sim \mathbb{CN}(0_{N_T}, \sigma_m^2 I_{N_T})$ denotes the additive white Gaussian noise (AWGN).

By introducing the ZF signal detection scheme cited at [27], the decoded signal can be represented as:

$$\bar{X}_0 = (H_0^H H_0)^{-1} H_0^H Y_0 \cong \sqrt{P_1} \left\{ W_{\bar{k}} \ddot{x}_{\bar{k}} + \sum_{\bar{j}=1, \bar{j} \neq \bar{k}}^{\bar{j}=8} W_{\bar{j}} \ddot{x}_{\bar{j}} \right\} \tag{22}$$

The signal model presented in Equation (22) is rescaled to make it compatible to a desirable UAV transmission power $P_{uav}$ of matrix size $N_T \times N_T$ by multiplying with a normalized signal power $P_2$, and the power scaled signal can be represented as:

$$\bar{\bar{X}}_0 = \sqrt{P_2} \sqrt{P_1} X_0 = \sqrt{P_1 P_2} \left\{ W_{\bar{k}} \ddot{x}_{\bar{k}} + \sum_{\bar{j}=1, \bar{j} \neq \bar{k}}^{\bar{j}=8} W_{\bar{j}} \ddot{x}_{\bar{j}} \right\} \tag{23}$$

In the first time phase, the main UAV transmits its signal $\bar{\bar{X}}_0$ to all cooperative UAVs and ground mMTC users. The signal received at the ground mMTC user $\bar{k}$ can be represented as:

$$Y_{\bar{k}}^1 = H_{\bar{k}} \sqrt{P_1 P_2} \left\{ W_{\bar{k}} \ddot{x}_{\bar{k}} + \sum_{\bar{j}=1, \bar{j} \neq \bar{k}}^{\bar{j}=8} W_{\bar{j}} \ddot{x}_{\bar{j}} \right\} + n_{\bar{k}}^1 \tag{24}$$

where $n_{\bar{k}}^1 \sim \mathbb{CN}(0_{N_R}, \sigma_{\bar{k}}^2 I_{N_R})$ denotes the additive white Gaussian noise (AWGN). The signal received at the two cooperative UAVs are given by:

$$Y_{01}^1 = H_{01} \sqrt{P_1 P_2} \left\{ W_{\bar{k}} \ddot{x}_{\bar{k}} + \sum_{\bar{j}=1, \bar{j} \neq \bar{k}}^{\bar{j}=8} W_{\bar{j}} \ddot{x}_{\bar{j}} \right\} + n_{01}^1 \tag{25a}$$

$$Y_{02}^1 = H_{02} \sqrt{P_1 P_2} \left\{ W_{\bar{k}} \ddot{x}_{\bar{k}} + \sum_{\bar{j}=1, \bar{j} \neq \bar{k}}^{\bar{j}=8} W_{\bar{j}} \ddot{x}_{\bar{j}} \right\} + n_{02}^1 \tag{25b}$$

where $n_{01}^1 \sim \mathbb{CN}(0_{N_T}, \sigma_{01}^2 I_{N_T})$ and $n_{02}^1 \sim \mathbb{CN}(0_{N_T}, \sigma_{02}^2 I_{N_T})$ denote the AWGN, and the superscript 1 on different variables indicates the first time phase. The amplifying gain in this system at the two cooperative UAVs can be represented as:

$$G1 = \sqrt{\frac{P_{uav}}{P_{uav} ||H_{01}||^2 + \sigma_{01}^2 I_{N_T}}} \tag{26a}$$

$$G2 = \sqrt{\frac{P_s}{P_{uav} ||H_{02}||^2 + \sigma_{02}^2 I_{N_T}}} \tag{26b}$$

In the second time phase, both the cooperative UAVs transmit its amplified signals to the ground mMTC users. The signal received at the ground mMTC user $\bar{k}$ from cooperative UAVs can be represented as:

$$
\begin{aligned}
Y_{\bar{k}1}^2 &= H_{1\bar{k}}G1Y_{01}^1 + n_{\bar{k}1}^2 \\
&= H_{1\bar{k}}G1\left[ H_{01}\sqrt{P_1P_2}\left\{ W_{\bar{k}}\ddot{x}_{\bar{k}} + \sum_{\bar{j}=1,\bar{j}\neq\bar{k}}^{\bar{j}=8} W_{\bar{j}}\ddot{x}_{\bar{j}} \right\} + n_{01}^1 \right] + n_{\bar{k}1}^2 \\
&= H_{1\bar{k}}G1H_{01}\sqrt{P_1P_2}\left\{ W_{\bar{k}}\ddot{x}_{\bar{k}} + \sum_{\bar{j}=1,\bar{j}\neq\bar{k}}^{\bar{j}=8} W_{\bar{j}}\ddot{x}_{\bar{j}} \right\} + n_{\bar{k}1g}^2
\end{aligned}
\tag{27a}
$$

$$
\begin{aligned}
Y_{\bar{k}2}^2 &= H_{2\bar{k}}G2Y_{02}^1 + n_{\bar{k}2}^2 \\
&= H_{2\bar{k}}G2\left[ H_{02}\sqrt{P_1P_2}\left\{ W_{\bar{k}}\ddot{x}_{\bar{k}} + \sum_{\bar{j}=1,\bar{j}\neq\bar{k}}^{\bar{j}=8} W_{\bar{j}}\ddot{x}_{\bar{j}} \right\} + n_{02}^1 \right] + n_{\bar{k}2}^2 \\
&= H_{2\bar{k}}G2H_{02}\sqrt{P_1P_2}\left\{ W_{\bar{k}}\ddot{x}_{\bar{k}} + \sum_{\bar{j}=1,\bar{j}\neq\bar{k}}^{\bar{j}=8} W_{\bar{j}}\ddot{x}_{\bar{j}} \right\} + n_{\bar{k}2g}^2
\end{aligned}
\tag{27b}
$$

where $n_{\bar{k}1g}^2 \sim \mathbb{CN}(0_{N_R}, \sigma_{k1g}^2 I_{N_R})$ and $n_{\bar{k}2g}^2 \sim \mathbb{CN}(0_{N_R}, \sigma_{k2g}^2 I_{N_R})$ are the effective AWGN in transmission from the first and second cooperative UAV, respectively.

In the second phase, the total signal received by the ground mMTC user $\bar{k}$ can be represented as:

$$
\begin{aligned}
Y_{\bar{k}}^2 &= Y_{\bar{k}1}^2 + Y_{\bar{k}2}^2 \\
&= (H_{1\bar{k}}G1H_{01} + H_{2\bar{k}}G2H_{02})\sqrt{P_1P_2}\left\{ W_{\bar{k}}\ddot{x}_{\bar{k}} + \sum_{\bar{j}=1,\bar{j}\neq\bar{k}}^{\bar{j}=8} W_{\bar{j}}\ddot{x}_{\bar{j}} \right\} + n_{\bar{k}}^2
\end{aligned}
\tag{28}
$$

where the superscript 2 on different variables indicates the second time phase. By combining the signals received at the ground mMTC user $\bar{k}$ in both phases [33], the following can be written:

$$
\begin{aligned}
Y_d^{\bar{k}} &\triangleq \begin{bmatrix} Y_{\bar{k}}^1 \\ Y_{\bar{k}}^2 \end{bmatrix} = \begin{bmatrix} H_{\bar{k}} \\ (H_{1\bar{k}}G1H_{01} + H_{2\bar{k}}G2H_{02}) \end{bmatrix} \sqrt{P_1P_2}\left\{ W_{\bar{k}}\ddot{x}_{\bar{k}} + \sum_{\bar{j}=1,\bar{j}\neq\bar{k}}^{\bar{j}=8} W_{\bar{j}}\ddot{x}_{\bar{j}} \right\} \begin{bmatrix} n_{\bar{k}}^1 \\ n_{\bar{k}}^2 \end{bmatrix} \\
&= \widehat{H}_{\bar{k}}\sqrt{P_1P_2}\left\{ W_{\bar{k}}\ddot{x}_{\bar{k}} + \sum_{\bar{j}=1,\bar{j}\neq\bar{k}}^{\bar{j}=8} W_{\bar{j}}\ddot{x}_{\bar{j}} \right\} + n_{\bar{k}}
\end{aligned}
\tag{29}
$$

where $\widehat{H}_{\bar{k}} = \begin{bmatrix} H_{\bar{k}} \\ (H_{1\bar{k}}G1H_{01} + H_{2\bar{k}}G2H_{02}) \end{bmatrix}$ is the effective channel between the UAVs and ground mMTC user $\bar{k}$ and is of $2N_R \times N_T$ matrix in size, while the combined signal $Y_d^{\bar{k}}$ and AWGN noise $n_{\bar{k}} \sim \mathbb{CN}(0_{2N_R}, \sigma_{\bar{k}}^2 I_{2N_R})$ are of $2N_R \times N_L$ matrix in size. On further simplification, Equation (29) can be re-written as:

$$
Y_d^{\bar{k}} = \widehat{\widehat{H}}_{\bar{k}}\left\{ W_{\bar{k}}\ddot{x}_{\bar{k}} + \sum_{\bar{j}=1,\bar{j}\neq\bar{k}}^{\bar{j}=8} W_{\bar{j}}\ddot{x}_{\bar{j}} \right\} + n_{\bar{k}}
\tag{30}
$$

where $\widehat{\widehat{H}}_{\bar{k}} = \widehat{H}_{\bar{k}}\sqrt{P_1P_2}$ is the effective channel for the signal model presented in Equation (30) and it is also of $2N_R \times N_T$ matrix in size. Generally, such non-symmetric complex channel matrix is found to be sparse in nature and due to sparsity of matrix, its pseudo inverse operation for achieving exact solution from signal model presented in Equation (30) cannot be obtained. Using the regularized ZF equalization technique [34] to invert the effect of the channel $\widehat{\widehat{H}}_{\bar{k}}$, the detected signal in receiver $\bar{s}_d^{\bar{k}}$ of $N_T \times N_L$ matrix in size can be represented as:

$$\bar{S}_d^{\bar{k}} = (\widehat{\bar{H}}_{\bar{k}}^H \widehat{\bar{H}}_{\bar{k}} + \mathfrak{H} I_{N_T})^{-1} \widehat{\bar{H}}_{\bar{k}}^H Y_d^{\bar{k}}$$

$$= (\widehat{\bar{H}}_{\bar{k}}^H \widehat{\bar{H}}_{\bar{k}} + \mathfrak{H} I_{N_T})^{-1} \widehat{\bar{H}}_{\bar{k}}^H \widehat{\bar{H}}_{\bar{k}} \left\{ W_{\bar{k}} \dddot{x}_{\bar{k}} + \sum_{\bar{j}=1, \bar{j} \neq \bar{k}}^{\bar{j}=8} W_{\bar{j}} \dddot{x}_{\bar{j}} \right\} + (\widehat{\bar{H}}_{\bar{k}}^H \widehat{\bar{H}}_{\bar{k}} + \mathfrak{H} I_{N_T})^{-1} \widehat{\bar{H}}_{\bar{k}}^H n_{\bar{k}}$$

$$= \left\{ W_{\bar{k}} \dddot{x}_{\bar{k}} + \sum_{\bar{j}=1, \bar{j} \neq \bar{k}}^{\bar{j}=8} W_{\bar{j}} \dddot{x}_{\bar{j}} \right\} + \hat{n}_{\bar{k}} \tag{31}$$

where $\mathfrak{H} = 1.0 \times 10^{-19}$ is the regularization parameter, and $\hat{n}_{\bar{k}}$ is the AWGN noise. From Equation (31), it is quite obvious that the received signal in ground mMTC user $\bar{k}$ contains its own and interference signals from other mMTC users. Multiplying Equation (31) by $H_{\bar{k}}$, a modified form of the received signal $\tilde{S}_d^{\bar{k}}$ of $N_R \times N_L$ matrix in size is obtained, and it can be represented as:

$$\tilde{S}_d^{\bar{k}} = H_{\bar{k}} \bar{S}_d^{\bar{k}}$$

$$= H_{\bar{k}} W_{\bar{k}} \dddot{x}_{\bar{k}} + H_{\bar{k}} \sum_{\bar{j}=1, \bar{j} \neq \bar{k}}^{\bar{j}=8} W_{\bar{j}} \dddot{x}_{\bar{j}} + \widehat{\hat{n}}_{\bar{k}} \tag{32}$$

In Equation (32), all the components are of $N_R \times N_L$ matrix in size, and the noise component $\widehat{\hat{n}}_{\bar{k}} \sim \mathbb{CN}(0_{N_R}, \widehat{\hat{\sigma}}_{\bar{k}}^2 I_{N_R})$ represents the AWGN noise.

Again, $||H_{\bar{k}} W_{\bar{k}}||^2$ represents the instantaneous total signal power in two consecutive time phases for the mMTC user $\bar{k}$, the total instantaneous interference power of the signal received by mMTC user $\bar{k}$ for the $\bar{j}$-th mMTC user is represented by $||H_{\bar{k}} W_{\bar{k}}||^2$, and $N_R \widehat{\hat{\sigma}}_{\bar{k}}^2$ denotes the AWGN noise power. The received signal-to-interference-plus-noise ratio (SINR) for the mMTC user $\bar{k}(SINR_{\bar{k}})$ after gyre decoding can be represented as:

$$SINR_{\bar{k}} = \frac{||\bar{H}_{\bar{k}} W_{\bar{k}}||^2}{\sum_{\bar{j}=1, \bar{j} \neq \bar{k}}^{\bar{j}=8} ||\bar{H}_{\bar{k}} W_{\bar{k}}||^2 + N_R \widehat{\hat{\sigma}}_{\bar{k}}^2} \tag{33}$$

The logarithmic function of SINR represents the achievable ergodic rate $R_{\bar{k}}$ for mMTC user $\bar{k}$, and it can be written as after gyre decoding [35]:

$$R_{\bar{k}} = \mathbb{E}\{\log_2(1 + SINR_{\bar{k}})\}$$

$$= \mathbb{E}\left\{ \log_2 \left( 1 + \frac{||\bar{H}_{\bar{k}} W_{\bar{k}}||^2}{\sum_{\bar{j}=1, \bar{j} \neq \bar{k}}^{\bar{j}=8} ||\bar{H}_{\bar{k}} W_{\bar{k}}||^2 + N_R \widehat{\hat{\sigma}}_{\bar{k}}^2} \right) \right\} \tag{34}$$

However, from the signal model presented in Equation (32), it is quite obvious that the reduction in the interfering signal power is very much significant as the GP removes MUI by designing GP beamforming weight $W_{\bar{k}}$ assigned for mMTC user $\bar{k}$ to fall in the null space of the total MIMO flat-fading channel matrix $H$ and produces $\bar{H}_{\bar{k}} W_{\bar{j}} = 0$. Considering $H_{\bar{k}} W_{\bar{k}} = \bar{H}_{\bar{k}}$ and multiplying Equation (32) $(\bar{H}_{\bar{k}}^H \bar{H}_{\bar{k}})^{-1} \bar{H}_{\bar{k}}^H$, a new form of the processed signal for mMTC user $\bar{k}$ is obtained as:

$$\dddot{x}_{\bar{k}} = (\bar{H}_{\bar{k}}^H \bar{H}_{\bar{k}})^{-1} \bar{H}_{\bar{k}}^H \bar{H}_{\bar{k}} \dddot{x}_k + (\bar{H}_{\bar{k}}^H \bar{H}_{\bar{k}})^{-1} \bar{H}_{\bar{k}}^H H_{\bar{k}} \sum_{\bar{j}=1, \bar{j} \neq \bar{k}}^{\bar{j}=8} W_{\bar{j}} \dddot{x}_{\bar{j}} + (\bar{H}_{\bar{k}}^H \bar{H}_{\bar{k}})^{-1} \bar{H}_{\bar{k}}^H \widehat{\hat{n}}_{\bar{k}}$$

$$= \dddot{x}_{\bar{k}} + (\bar{H}_{\bar{k}}^H \bar{H}_{\bar{k}})^{-1} \bar{H}_{\bar{k}}^H H_{\bar{k}} \sum_{\bar{j}=1, \bar{j} \neq \bar{k}}^{\bar{j}=8} W_{\bar{j}} \dddot{x}_{\bar{j}} + \widehat{n} \tag{35}$$

where the first, second, and third terms of Equation (35) are indicative of signal, interference, and noise components $\widehat{n} = (\bar{H}_{\bar{k}}^H \bar{H}_{\bar{k}})^{-1} \bar{H}_{\bar{k}}^H \widehat{\hat{n}}_{\bar{k}}$ and is $\sim \mathbb{CN}(0_{N_R}, \widehat{\sigma}_{\bar{k}} I_{N_R})$, respectively.

With further processing for necessary steps with the execution of the inverse T-transformation, achievable ergodic rate, SINR, and SNR values can be estimated.

## 3. Numerical Results and Relative Discussion

This section presents the numerical outcomes for the secured mmWave cooperative UAV-aided CP-less GFDM system. The considered simulation parameters are presented in Table 1. Based on the noise power of $-80$ dBm, which corresponds to $1 \times 10^{-11}$ W, the Additive white Gaussian noise (AWGN) component is introduced at different receive antennas of the main UAV [36]. In this scenario, there is a ground BS, one main and two cooperative UAVs, which are connected in an integrated UAV-terrestrial network to serve eight massive machine-type communications (mMTC) users, which are 1.5 m above the ground surface. The heights of the main and cooperative two UAVs are 120 m, 100 m, and 100 m, respectively. The transmitting antenna height of the ground BS is 50 m, and the 3D transmission distances from the ground BS to the main UAV, first cooperative UAV, and second cooperative UAV are 157.80 m, 122.47 m, and 187.0829 m, respectively, and from the main UAV to eight mMTC users are 131.31 m, 128.62 m, 123.66 m, 121.28 m, 124.77 m, 127.64 m, 133.22 m, and 122.41 m, respectively. The 3D transmission distance from the main UAV to the two cooperative UAVs is identical and is of 53.85 m. Considering the heights of the buildings and ground base station, the cooperative UAV's trajectory path is preferably being considered as circular and not random. Thus, the chance of any conflict with existing infrastructure is absolutely minimal. Three-dimensional Brownian perturbation model-based [30] trajectory planning combines the randomness with directed motion, and thus, the role of UAVs in successful signal transmission is being observed.

**Table 1.** Simulation Parameters.

| Description | Value |
| --- | --- |
| No. of bits | 2048 |
| No. of subcarriers | 256 |
| No. of sub-symbols in each GFDM block | 10 |
| Time of each sub-symbol | 66.67 |
| No. of GFDM blocks | 6 |
| Carrier spacing (KHz) | 15 |
| GFDM signal bandwidth (MHz) | 3.84 |
| UAV transmission power | 1–10 W |
| Signal-to-noise ratio, $E_b/N_0$ (dB) | 0–30 |
| Noise power (dBm) | $-80$ |
| Average transmission power/channel | 35.68 dBm (3.70 W) |

The energy consumption of each UAV associates both communication-related energy consumption and propulsion energy consumption. Here, a single UAV controlling ground node (GN) based energy-efficient rotary-wing UAV communications is considered, and a 3D-based Brownian perturbation mobility model is applied, which controls UAV speed/deviation from the UAV target position, which, in turn, reduces propulsion energy consumption [37]. Basically, the proposed scheme can importantly balance the additional consumption of power of UAVs. The transmitting UAVs and ground mMTC users have different channel state information (CSI). In such case, MIMO flat-fading channels are estimated utilizing the probabilistic path loss model cited in [32].

In this study, the implementation of the PLS technique is based on the utilization of Lorenz's hyperchaos mapping system presented in Figure 3. Chaotic systems have been extensively used in physical layer encryption because of their high initial sensitivity and good randomness. In chaotic encryption schemes, sharing of the initial value between the transmitter and receiver is an important factor of system security, which makes it difficult for illegal users to steal information through malicious attacks. To achieve higher security performance, Lorenz's hyperchaos mapping system is considered by introducing new state variables in the encryption.

Figure 3 shows that the behavior of the system is fully dependent on the considered parameter values. In Figure 4, a 3D plot of various keys assigned to individual mMTC user is shown, which confirms that assigned keys are different for different mMTC users.

In Figure 5, the cumulative distribution function (CDF) of the 3D positioning errors for both the main and two cooperative UAVs have been presented in a static scene. It can be seen from the CDF that the error at 95% confidence is 0.08 m in vertical Z direction and average 0.06 m in both horizontal X and Y directions for the main UAV. In the case of the cooperative UAV #1 with identical confidence level, the positional error is 0.06 m in vertical Z direction and 0.12 m in horizontal X direction and 0.05 m in horizontal Y direction. In the case of cooperative UAV #2, the positional error is 0.06 m in

vertical Z direction and 0.02 m in horizontal X direction and 0.04 m in horizontal Y direction. In all cases, the 3D positional errors of the UAVs are reasonably acceptable in the context of state-of-the-art.

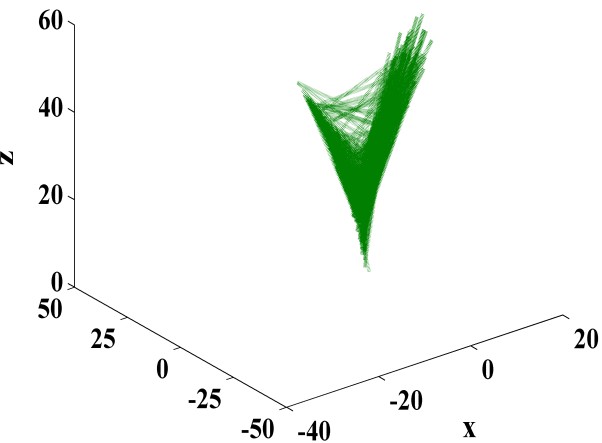

**Figure 3.** Three-dimensional reconstructed attractor of Lorenz's hyperchaos mapping system in the $(x, y, z)$-plane.

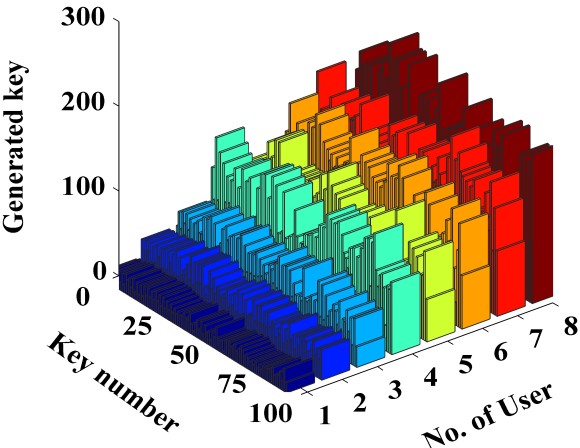

**Figure 4.** Generated keys for using Lorenz's hyperchaos mapping system for different mMTC users.

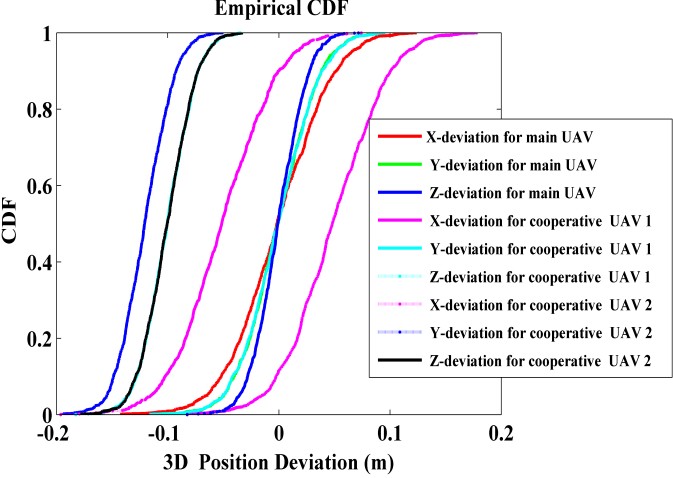

**Figure 5.** Empirical CDFs $x, y, z$ values relative to the UAV target position.

The impact of both gyre decoding and the T-transformation scheme on the received signal-to-noise ratio (SNR), received SINR, and achievable ergodic rate with the variation in UAV transmission

power for eight mMTC users have been illustrated through Figures 6–8. In Figure 6a, the estimated average received SNR values for eight mMTC users without the application of gyre decoding and inverse T-transformation schemes are presented. From Figure 6a, it is seen by analyzing all the estimated SNR values for mMTC user 1 through mMTC user 8, the maximum average SNR value is approximately 30.533 dB, and the minimum average SNR value is approximately 20.529 dB. In Figure 6b, the estimated average SNR values for eight mMTC users with the implementation of both gyre decoding and inverse T-transformation schemes at the receiver are presented. There is no significant difference observed while comparing with the SNR values in the absence of gyre decoding and T-transformation schemes, as shown in Figure 6a. It is seen from Figure 6b by estimating all SNR values for mMTC user 1 through mMTC user 8 that the maximum average SNR value is approximately 29.663 dB, and the minimum average SNR value is approximately 19.659 dB.

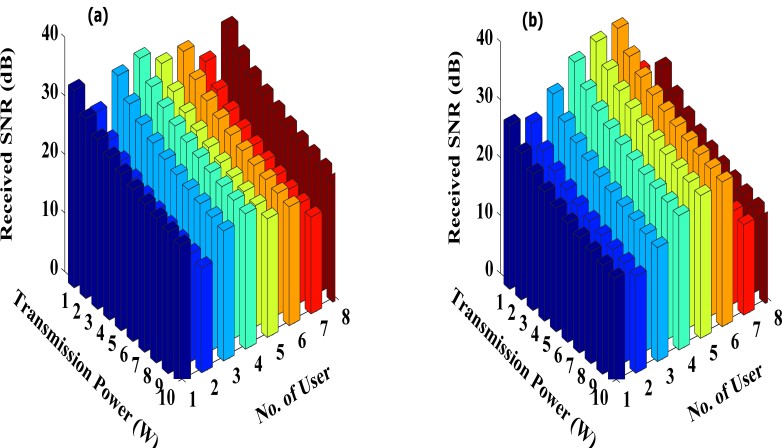

**Figure 6.** Estimated average received signal-to-noise ratio (SNR) when both gyre decoding and inverse T-transformation schemes are (**a**) absent and (**b**) present.

In Figure 7a, it is quite clear that due to the presence of interfering signals, the effective received SINR values for each mMTC user are reduced. The estimated SINR values of each mMTC user are almost constant in spite of the variation of UAV transmission power. The estimated SINR values for mMTC user 1 through mMTC user 8 are almost −4.8310 dB, −9.0268 dB, −6.9553 dB, −6.4912 dB, −10.2783 dB, −9.8867 dB, −12.7168 dB, and −11.8770 dB, respectively. Figure 7b shows that due to the simultaneous implementation of both gyre decoding and inverse T-transformation schemes based orthogonal spreading codes, the interfering signal power is totally nullified, causing improved SINR performance as compared to the results presented in Figure 7a. The estimated SINR values for mMTC user 1 through mMTC user 8 are identical to SNR values presented in Figure 6b.

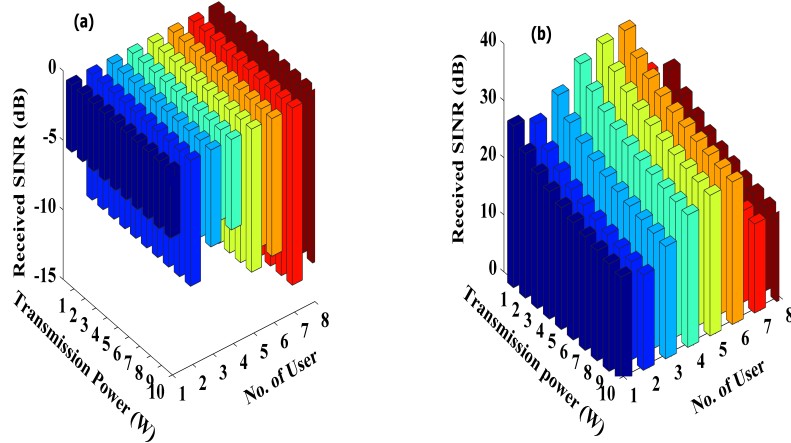

**Figure 7.** Estimated average received signal to interference noise ratio (SINR) when both gyre decoding and inverse T-transformation schemes are (**a**) absent and (**b**) present.

In Figure 8a, it is observable that even when the UAV transmission power is varied, the estimated achievable ergodic rate for different mMTC users is comparatively low and almost constant. The estimated achievable ergodic rate for mMTC user 1 through mMTC user 8 are 1.2345 bps/Hz, 0.5120 bps/Hz, 0.7976 bps/Hz, 0.8790 bps/Hz, 0.3894 bps/Hz, 0.4243 bps/Hz, 0.2263 bps/Hz, and 0.2731 bps/Hz, respectively. In Figure 8b, it is observable that the estimated achievable ergodic rates for different mMTC users are very much improved with the application of gyre decoding and T-transformation scheme than in the absence of them, as shown in Figure 8a. It is also noticeable from the figure that the achievable ergodic rate values decrease when the UAV transmission power is varied. By analyzing all the estimated achievable ergodic rates for mMTC user 1 through mMTC user 8, the maximum and minimum average achievable ergodic rate is found to have values 9.856 bps/Hz and 6.553 bps/Hz, respectively.

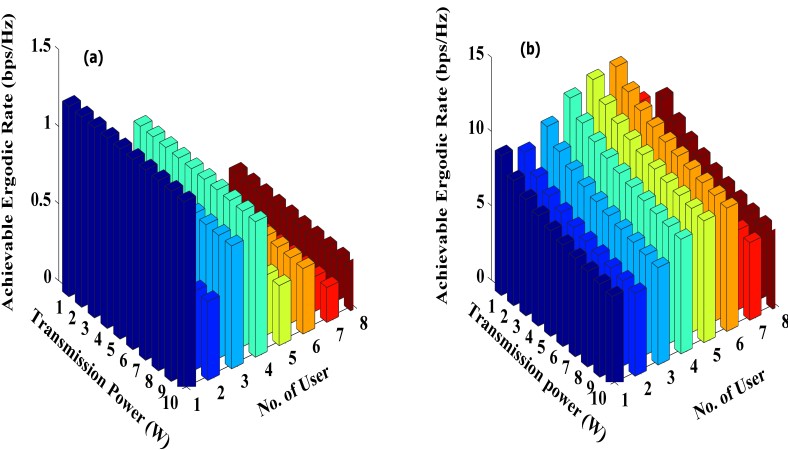

**Figure 8.** Estimated achievable ergodic rate when both gyre decoding and inverse T-transformation schemes are (**a**) absent and (**b**) present.

Considering Brownian motion, it is observable that for the $x$, $y$, $z$ coordinate variation of 0.2 m relative to UAV's fixed positions, the mean error in estimated achievable ergodic rates at different $E_b/N_0$ values is found to be of 0.26 bps/Hz. On the other hand, in the case of $x$, $y$, $z$ coordinate variation of $-0.2$ m relative to UAV's fixed positions, the mean error in estimated achievable ergodic rates is of $-0.83$ bps/Hz. In Figure 9, the estimated achievable ergodic rate values for ground mMTC users confined within a circular region of a 50 m radius are presented. In such case, ground BS transmit power (46 dBm = 39.81 W) and UAV base station transmit power (30 dBm = 1 W) from Table 10.3 of [5] has been considered. From the 3D displayed achievable ergodic rate of Figure 9 with identical heights of all the UAVs at 100 m, 200 m, and 300 m, it is seen that with increase in UAV height, the estimated achievable ergodic rate values decrease. The maximum estimated achievable ergodic rate in the case of UAV heights of 100 m, 200 m, and 300 m are found to have values of 14.26 bps/Hz, 13.19 bps/Hz, and 12.21 bps/Hz.

In Figure 10, estimated maximum achievable ergodic rate values at different normalized signal-to-noise ratios $(E_b/N_0)$ are compared with the similar works of different authors from [38–40]. It is quite clear from the figure that the proposed system outperforms all the other similar works utilizing different systems. Performance in terms of achievable ergodic rate is even better with gyre decoding and T-transformation scheme utilized in tandem than merely with gyre decoding.

An acceptable out-of-band (OOB) power reductions of 318.77 dB, 317.31 dB, 319.25 dB, 321.71 dB, 320.93 dB, 321.22 dB, 320.73 dB, and 319.97 dB are attained for the case of mMTC user 1 through mMTC user 8, respectively. The result illustrated in Figure 11 is a typical case of OOB power reduction performance of mMTC user 1.

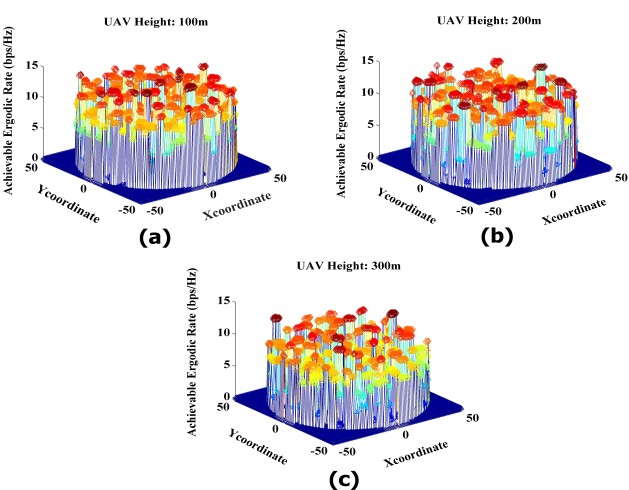

**Figure 9.** Estimation of achievable ergodic rate of randomly distributed mMTC users within a circular region of a 50 m radius and all UAVs at heights: (**a**) 100 m, (**b**) 200 m, and (**c**) 300 m.

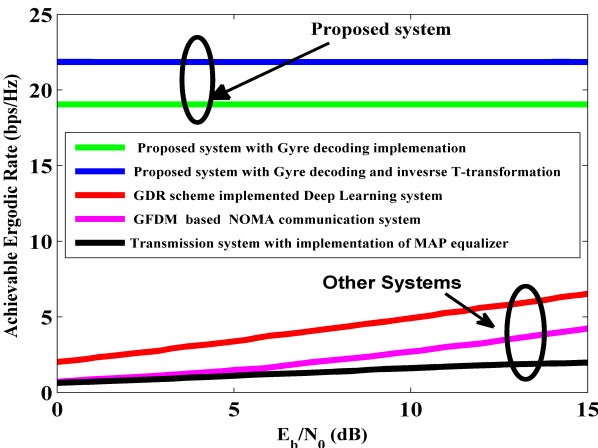

**Figure 10.** Comparative analysis of achievable ergodic rates between the proposed system and other similar communication systems.

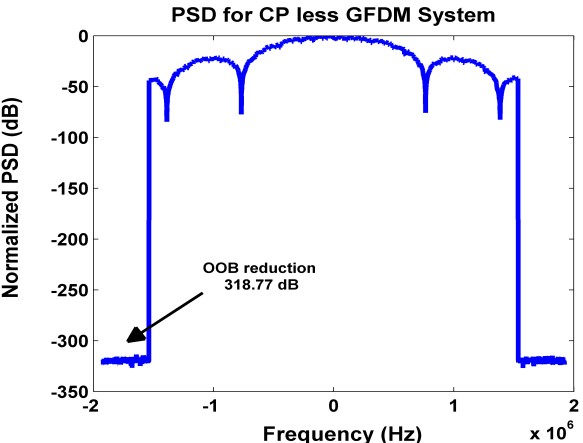

**Figure 11.** Power spectral density of the proposed system for typically chosen case of mMTC user 1.

To illustrate the efficacy of the proposed system further, BER performance is also analyzed by implementing concatenated channel coding with multi-user beamforming weighting-aided maximum-likelihood and zero forcing (ZF) detection technique adopting 16-QAM digital modulation. It is quite easily remarkable from the simulation outcomes presented in Figures 12–14 that the BER performance

of the proposed system implementing (3,2) single parity check (SPC) and repeat and accumulate (RA) concatenated channel-coding scheme is better than other two concatenated channel-coding schemes tested in this system.

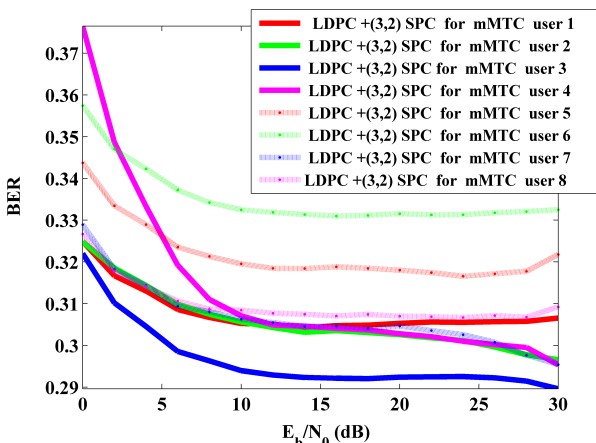

**Figure 12.** BER analysis of the proposed system adopting LDPC and (3,2) SPC concatenated channel-coding technique.

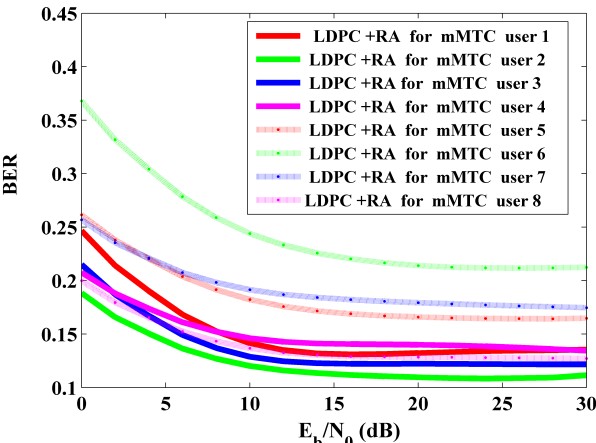

**Figure 13.** BER analysis of the proposed system adopting LDPC and RA concatenated channel-coding technique.

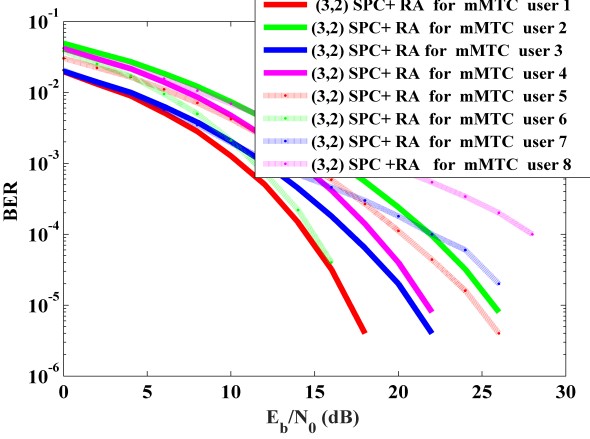

**Figure 14.** BER analysis of the proposed system adopting (3,2) SPC and RA concatenated channel-coding technique.

It is seen from Figure 12 that the BER varies from 33.30% to 29.40% for a typically assumed $E_b/N_0$ value of 10 dB with the system utilizing low density parity check (LDPC) and (3,2) SPC concatenated channel-coding technique. Overall, BER performance is not quite satisfactory for any of the mMTC users. In SNR region of 11 dB to 21 dB, some mMTC users are getting their signals with almost identical BER.

In Figure 13, comparatively improved system performance is noticeable with the system tested under the LDPC and RA concatenated channel-coding technique. It is seen by observing simulated data of all eight users in Figure 13 that the BER varies from approximately 24.39% to 12.00% for a typically granted $E_b/N_0$ value of 10 dB.

In Figure 14, BER performance of the system under concatenated channel-coding scheme with a combination of (3,2) SPC and RA is quite satisfactory. Over the wide range of SNR from 0 dB to 30 dB, the estimated maximum and minimum BER are found to have values 0.14190 (14.19%) and 0 (0%), respectively. It is seen from Figure 14 that the BER varies from 2.75% to 0.51% for a typically assumed $E_b/N_0$ value of 10 dB. It is also observed that in the case of mMTC user 1, BER is reduced from 5.90% to 0.68% with an increase in the SNR value from 6 dB to 16 dB.

In Figure 15, BER performance of the proposed system utilizing (3,2) SPC and RA concatenated channel-coding technique has been compared with other GFDM systems introduced in previous works. From a brief description perspective of the systems presented in [39,41–43], it can be said that the authors in [41] did a thorough analysis on the applicability of wireless energy harvested GFDM-based cooperative network and presented numerical results in terms of BER. In [42], the effectiveness of localized discrete gabor transform (LDGT) algorithm implemented the GFDM system with varying windowing lengths was studied accounting the BER performance. Authors in [39] conducted a performance evaluative study for a GFDM-assisted NOMA system with respect to BER and achievable rate analysis. In [43], the authors highlighted the effectiveness of a designed pulse shaping filter based on quadratic programming in reducing the OOB radiation and BER performance of the GFDM system. Now, it is quite clearly visible from Figure 15 that the proposed system is showing comparatively better performance in terms of BER against $E_b/N_0$ than cooperative GFDM, LDGT-aided GFDM, GFDM-based NOMA, and designed pulse shaping filter implemented GFDM systems.

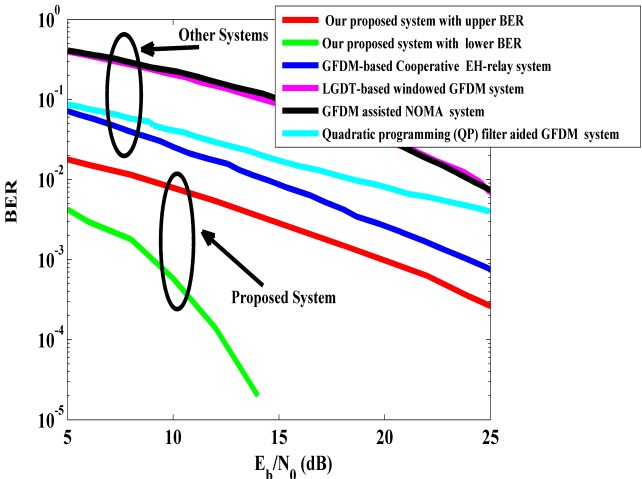

**Figure 15.** Comparative BER performance analysis between the proposed system and other B5G compatible multicarrier systems.

In Figure 16, it is clear from the high BER values that the improved 3D controlled Lorenz mapping system-aided PLS encryption technique is very much effective in offering secure multi-user data transmission. In addition, it is difficult to identify the transmitted data correctly at the receiver without any knowledge of assigned parameter values used in such an encryption technique.

Due to the utilization of a channel-dependent precoding technique in this proposed system, the power of different signals at different transmitting channels is varied. As a consequence, PAPR values also vary. It is observable from Figure 17 that at a complementary CDF (CCDF) of $10^{-3}$, the maximum and minimum PAPR values are 9.5 dB (Transmitting antenna 2) and 8 dB (Transmitting antenna 4), respectively. As it is seen that the variation among all the transmitting channels is well within the PAPR value of 1.5 dB, curves for different channels coincides at some stage.

In Figure 18, it is observable that the estimated PAPR values at different ground transmitting channels in the proposed system are reasonably acceptable if they are considered with respect to the state-of-the-art.

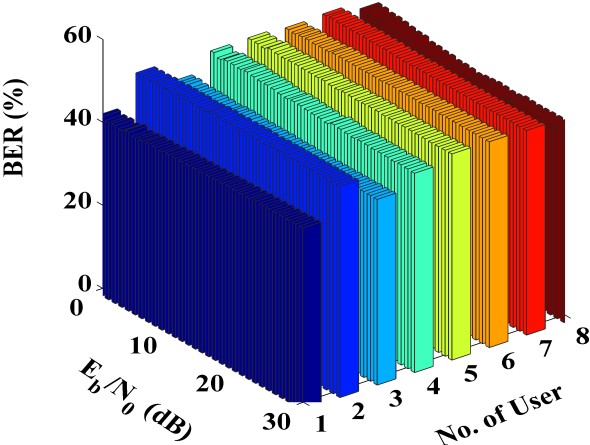

**Figure 16.** BER performance for the different mMTC users without improved 3D controlled Lorenz mapping system-aided PLS decryption scheme.

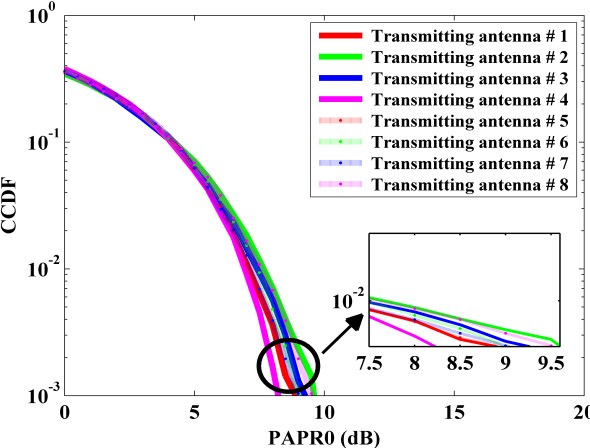

**Figure 17.** Estimated complementary cumulative distribution function (CCDF) of peak-to-average power ratio (PAPR) at different ground transmitting channels of the proposed simulated system.

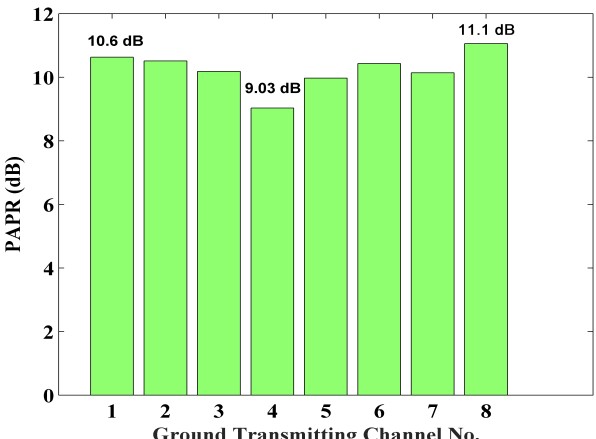

**Figure 18.** Estimated peak-to-average power ratio (PAPR) at different ground transmitting channels of the proposed simulated system.

## 4. Conclusions

A UAV-aided mmWave downlink cooperative CP-less GFDM system is proposed in this paper. UAVs are combined with a terrestrial cellular network, and emphasis is placed on physical layer security for massive machine-type communication (mMTC) users in such a network. In this study, the implementation of an efficient low-complexity T-transformation spreading code as well as MU-MIMO Gyre precoding resulted in the reduction in both MUI and the enhancement of the achievable ergodic rate. Utilizing null subcarriers at both ends of a GFDM block for sub-time symbols and better than raised cosine (BTRC) filter resulted in significant reduction in out-of-band (OOB) spectrum power by eliminating the effects of the fading channel at mmWave. The 3D mobility model resulted in reducing the effects of the Brownian motion of UAVs. In this proposed system, the performance in terms of BER is not quite satisfactory. However, the BER performance is reasonably acceptable with the combined implementation of (3,2) SPC and RA channel-coding schemes and outperforms other proposals found in the literature. In the future, introducing massive MIMO to the proposed system can improved it and make it more robust.

**Author Contributions:** Conceptualization, J.J.S., S.E.U. and R.R.; Data curation, J.J.S., S.E.U., R.R. and M.R.I.; Formal analysis, J.J.S., S.E.U., R.R., M.R.I. and M.M.R.; Funding acquisition, R.R., M.R.I. and M.A.P.M.; Investigation, J.J.S., S.E.U., R.R. and M.M.R.; Methodology, J.J.S. and S.E.U.; Project administration, S.E.U., R.R., M.R.I. and A.Z.K.; Resources, J.J.S. and S.E.U.; Software, J.J.S. and S.E.U.; Supervision, S.E.U., R.R., M.R.I. and A.Z.K.; Validation, R.R., M.R.I. and M.M.R.; Visualization, J.J.S., M.M.R. and M.A.P.M.; Writing—original draft, J.J.S. and S.E.U.; Writing—review & editing, R.R., M.R.I., M.M.R., A.Z.K. and M.A.P.M. All authors have read and agreed to the published version of the manuscript.

**Funding:** This research received no external funding.

**Conflicts of Interest:** The authors declare no conflict of interest.

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
