# Peer review of "Gyre Precoding and T-Transformation-Based GFDM System for UAV-Aided mMTC Network"

_electronics, doi:10.3390/electronics10232915_

Round 1
Reviewer 1 Report
This paper investigates an unmanned aerial vehicle (UAV)-aided multi-antenna configured downlink mmWave cooperative generalized frequency division multiplexing (GFDM) system. Basically, it is well written. However, the reviewer has some minor concerns as follows:
- In this manuscript, punctuation should be added after each expression.
- (10) gives an air-to-ground LoS probability model. Please add references to explain the rationality of the model.
- Some simulation diagrams need to be improved. For example, in Figure 20, since the curves are approximately coincident, a small window should be added to enlarge these curves and explain the reason for the coinciding.
- Is the setting of simulation parameters such as noise power practical? Please provide more references to support the assumption.
- There are some typo and grammatical errors that need to be corrected.
In this work, the authors studied the PLS for mMTC users problem in a UAV-aided multi-antenna configured downlink mmWave cooperative GFDM system. Some techiques are employed to improve the system’s performance, such as using T-transformation spreading codes and MU-MIMO Gyre precoding to reduce MUI and improve ergodic rate.
In general, the paper is well written and explores an interesting problem. But I have the following concerns:
- The literature review should be enhanced. The introduction part is not easy to follow, e.g., more explanations are needed to introduce PLS and why Lorenz’s hyperchaos mapping system is used in this paper.
- The authors listed too many contributions, I suggest reducing the number and re-suming the main contributions. For example, the second paragraph ' Null sub-symbols are…' may be a means of system implementation and should not be listed as a major contribution.
- In the simulation results, the advantages or performance gains of using Lorenz’s hyperchaos mapping system are worth discussing.
- In the simulation results, combining some figures together for analysis (e.g., combining Fig. 7 and Fig. 10 into Fig. 7a and Fig. 7b) would help the reader for easier understanding of the paper.
- In this manuscript, punctuation should be added after each expression.
- (10) gives an air-to-ground LoS probability model. Please add references to explain the rationality of the model.
- Some simulation diagrams need to be improved. For example, in Figure 20, since the curves are approximately coincident, a small window should be added to enlarge these curves and explain the reason for the coinciding.
- Is the setting of simulation parameters such as noise power practical? Please provide more references to support the assumption.
- There are some typo and grammatical errors that need to be corrected.
Reviewer 2 Report
This paper proposed a UAV-aided multi-antenna configured down-link mmWave cooperative GFDM system. Numerical and simulation results are presented to showcase the robustness of the system in terms of PLS, rate, and SINR. In general, the topic under investigation is interesting and timely. I have the following comments:
- The UAVs have a huge potential for deployment in future networks under various scenarios. The authors can refer to the following survey paper to highlight more use cases and challenges in the Introduction. [R1] M. M. Azari et al., “Evolution of non-terrestrial networks from 5G to 6G: A survey,” arXiv preprint arXiv:2107.06881, 2021.
- The propulsion energy consumption seems to be a major barrier in realising the UAV-based networks. The authors should discuss this issue.
- Multi-antenna systems may require large RF chains and thus corresponding ADCs may incur heavy payload which in turn, might affect the operational life cycle of the UAV. How does the multi-antenna being mounted on UAV affect the system performance should be discussed.
- Why the UAVs trajectory are considered to be random, does it make sense to let UAVs fly in open-air spaces without control? I think proper trajectory planning is needed for UAVs' operation in practice to avoid any conflict with existing infrastructure. Please comment on this as well.
- The authors are requested to address the above-raised points and discuss them appropriately in the revised manuscript.
Round 2
Reviewer 1 Report
Thanks for the efforts. All the reviewer's comments have been properly addressed. No more comments on the version.